# Maximizing Local Rewards on Multi-Agent Quantum Games through Gradient-Based Learning Strategies

**DOI:** 10.3390/e25111484

**Published:** 2023-10-26

**Authors:** Agustin Silva, Omar Gustavo Zabaleta, Constancio Miguel Arizmendi

**Affiliations:** ICYTE (Instituto de Investigaciones Científicas y Tecnológicas), Mar del Plata B7600, Argentina

**Keywords:** quantum computing, game theory, reinforcement learning

## Abstract

This article delves into the complex world of quantum games in multi-agent settings, proposing a model wherein agents utilize gradient-based strategies to optimize local rewards. A learning model is introduced to focus on the learning efficacy of agents in various games and the impact of quantum circuit noise on the performance of the algorithm. The research uncovers a non-trivial relationship between quantum circuit noise and algorithm performance. While generally an increase in quantum noise leads to performance decline, we show that low noise can unexpectedly enhance performance in games with large numbers of agents under some specific circumstances. This insight not only bears theoretical interest, but also might have practical implications given the inherent limitations of contemporary noisy intermediate-scale quantum (NISQ) computers. The results presented in this paper offer new perspectives on quantum games and enrich our understanding of the interplay between multi-agent learning and quantum computation. Both challenges and opportunities are highlighted, suggesting promising directions for future research in the intersection of quantum computing, game theory and reinforcement learning.

## 1. Introduction

As the vanguard of computational theory and machine learning increasingly intersects with the intricacies of quantum mechanics, a profound shift in our understanding of complex systems is emerging. At this crossroads lies the study of quantum games and multi-agent learning, a field poised to revolutionize myriad sectors through the marriage of quantum computing, game theory and reinforcement learning. This research elucidates this profound transformation by delving into the intricate dynamics of quantum games involving multiple learning agents.

Game theory, a cornerstone of modern economics and numerous branches of social sciences, provides an insightful mathematical framework for analyzing strategic scenarios and deciphering decision-making complexities in interactive situations. The framework of game theory, first articulated by von Neumann and Morgenstern [1], has become a wide array of applications, from conflict resolution to artificial intelligence. However, when quantum effects are incorporated, an even more fascinating framework emerges, quantum game theory. Quantum games, a generalization of classical games [2,3], offer a richer strategy space and more diverse outcomes due to quantum superposition and entanglement, allowing significant advantages over their classical counterparts. Quantum effects facilitate novel strategic behaviors that enable players to coordinate their strategies in unprecedented ways. These benefits have not gone unnoticed and there have been successful experimental realizations of Quantum Games [4,5], further highlighting their potential.

It is critical to emphasize that Nash proved [6] that all classical games with a finite set of strategies have a Nash equilibrium. This equilibrium represents a collection of strategies, one for each player, in which no player can enhance their outcome by independently altering their strategy. This also extends to an ε-approximate Nash equilibrium, in which no player can improve their outcome by more than ε through a unilateral change of their strategy. Similarly, it has been demonstrated that a Nash equilibrium will always exist in any quantum game [7], despite the fact that the set of strategies is no longer finite. It has also been proven that finding Nash equilibria in classical games is PPAD-complete and that finding ε-approximate Nash equilibria in both classical and quantum games is also in PPAD [8]. This means that there are no efficient (polynomial) algorithms for solving these problems. As a result, there is significant interest in designing heuristic algorithms that can find equilibria in these scenarios.

Meanwhile, Multi-Agent Reinforcement Learning (MARL) is a field concerned with how autonomous agents can learn and adapt to their environment to optimize their collective behavior. MARL, paired with game theory, has driven advances in autonomous driving [9], drone field coverage [10] and smart grid optimization [11]. Simultaneously, quantum game theory has begun to produce rich applications, including communications, where it has been applied to address issues of spectrum scarcity [12] and congestion in data networks [13] and economics, with quantum market games [14] and monetary economics benefiting [15], heralding a new era of quantum-enhanced solutions.

Recent articles have ventured into Quantum Machine Learning [16] and Quantum Reinforcement Learning [17,18,19] and, more related to our work, Quantum Multi-Agent Reinforcement Learning using Simulated Quantum Annealing [20], Quantum Botzmann Machines [21] and Variational Quantum Circuits [22]. However, the focus predominantly remains on quantum agents interacting within a classical environment. Although these models are highly valuable and can achieve good generalization with limited training data [23], they do not consider the challenges and opportunities posed by quantum environments.

The present discussion deviates from this more common route, probing the behavior of classical agents within quantum games. There are precedents that examine classical multi-agent learning within quantum games. Ref. [24] presents the evolution of quantum agents in a two-dimensional lattice, emphasizing their advantage over classical agents. The study is restricted to scenarios with three to five agents due to the lattice structure, limiting on three quantum gates from the whole SU(2) space as strategies in the Prisoner’s Dilemma and two specific quantum strategies in the Parrondo Paradox. Refs. [25,26] further explore quantum games in a two-dimensional lattice with cellular automata interactions, analyzing the prisoners’ dilemma and Battle of the Sexes, respectively. These studies allow players to use the complete set of strategies within the SU(2) space, but focusing only on two-player games. Ref. [26] also provides an in-depth analysis of the Battle of the Sexes game with incomplete information, addressing the uncertainties players have about the intentions of others. Ref. [27] investigates multi-agent quantum games’ dynamics, studying the evolution of agents in the Prisoner’s Dilemma, Snowdrift and Stag-Hunt games on evolving random networks. The analysis highlights the transitions between strategies in multi-agent games on evolving networks, providing insights into strategy dominance in various game scenarios but limiting games to two-player scenarios and each agent’s strategy space to only four possible quantum strategies (C, D, H, Q) from the SU(2) space. Finally, ref. [28] introduces a novel method for learning and visualizing strategies in quantum games. The authors propose a decentralized learning algorithm for two players, unaware of the game’s payoff matrix and the actions and rewards of the competitors. The study explores the impact of entanglement and noise on various quantum games’ performance. The algorithm allows agents to learn from the complete SU(2) space, even allowing for mixed strategies. Our work differs from the previous ones since it presents a gradient-based (versus gradient-free methods) algorithm applicable to general-sum quantum games that involve any number of players, which can select quantum strategies from the complete SU(2) space.

The core of our study introduces an algorithm in which agents selfishly and independently utilize the Adam algorithm [29] to adjust their actions, a trio of continuous real values. This policy-based and gradient-descent approach provides flexibility by allowing agents to directly learn actions from rewards, ensures continuous policy improvements and results in locally optimized strategies. Our results reveal the remarkable efficacy of this algorithm across various game configurations, including games involving two to five players, which could also be extended to *N* players. The learning agents display proficient adaptation and, notably, are able to easily and methodically highlight when the performance of agents playing in quantum games surpasses that in their classical counterparts.

In addition to exploring the learning dynamics in quantum games, this work delves into the effects of quantum noise on algorithm performance. Contrary to the common belief that quantum noise is often regarded as a performance impediment [30], our analysis presents an intriguing finding: under certain conditions, a slight amount of quantum noise can actually improve performance. Such a counterintuitive phenomenon is particularly relevant in the era of Noisy Intermediate-Scale Quantum (NISQ) computers [31], where noise is considered a fundamental part of the system.

The findings of this work enhance our understanding of quantum game dynamics and open avenues for practical applications. The successful performance of the quantum game learning algorithm and its uncovering of the counterintuitive potential of quantum noise to enhance system performance provide meaningful contributions to quantum computing, strategic decision-making and machine learning. By enabling more accurate modeling of complex interactions and improved optimization strategies, this research brings us a step closer to realizing the vast potential of quantum computing in real-world scenarios.

The remainder of this paper is organized as follows. Section 2 provides a detailed discussion of quantum games and the proposed multi-agent learning model. Section 3 develops the application of the algorithm across various game configurations and compares its performance with classical games. Section 4 explores the intriguing effects of quantum noise on the algorithm’s performance. Finally, Section 5 provides a comprehensive analysis of the insights obtained from the study and its potential applications.

## 2. Description of the Model

### 2.1. Classical and Quantum Games

The goal of game theory is to analyze decision-making systems involving two or more players cooperating or not with each other. An important feature in games is that the reward one player receives depends not only on the strategy she chooses, but also on other player strategies. It is well known that a game is defined by three elements: players, strategies and rewards. This work is based on N players’ games with two pure strategies each. The rewards are defined by a $2^{N}$-row (one for each combination of joint actions) payoff matrix where each row has N columns (one for each player), each column representing the reward for each player given a combination of the joint action. In Table 1, it is possible to observe a general representation of a two-player two-strategy payoff matrix where A0 and A1 are the strategies of player A; B0 and B1 are the strategies of player B; Ra1, Ra2, Ra3 and Ra4 are the rewards of player A; Rb1, Rb2, Rb3 and Rb4 are the rewards of player B. Consequently, a general payoff matrix for three and four players can be found in Table 2 and Table 3, respectively.

The following N-player two-strategy games are considered throughout the article: the minority problem [32], which rewards agents who select a strategy that was selected by 50% of players or less. The platonia dilemma [33], which rewards a player if she chooses strategy 1 and all other agents choose 0. The unscrupulous dilemma [34], which is an N-player extension of the well-known prisoners’ dilemma. Finally, the volunteers’ dilemma [35], which is an N-player extension of the famous chicken game. All these games will be further expanded on in Section 3. A three-player payoff matrix of these games will be shown as an example even though they are all extendable to N players: the minority problem in Table 4, the platonia dilemma in Table 5, the unscrupulous dilemma in Table 6 and the volunteers’ dilemma in Table 7.

To model quantum games, we followed the *EWL* [2] protocol for two players and then extended it to N players in [3]. The *first* step involves assigning a quantum state to each of the possible strategies. In the case of two strategies, for example, in the prisoners’ dilemma, *cooperate* →|0〉 and *defect*→|1〉. The *second* step is to create a quantum circuit where each player is assigned a qubit that starts in state |0〉. The *third* step is to create an entangled state between all players. This is done by applying the entangling operator $J=cos\left(\frac{\gamma }{2}\right)*I^{⊗N}+i*sin\left(\frac{\gamma }{2}\right)*\sigma_{x}^{⊗N}$, as seen in Figure 1 for N = 2 players, where I is the identity matrix, $\sigma_{x}$ the Pauli X-gate, *N* represents the number of players and γ a value that determines the amount of entanglement, γ=0 being not entanglement at all (classical games) and $\gamma =\frac{\pi }{2}$ maximum entanglement (fully quantum games).

In the *fourth* step, every player chooses her most suitable strategy individually and independently. This is done by modifying the state of her own qubit locally. To do this, every player applies one or more one-qubit gates, modifying the state of her qubit. A general one-qubit gate [36] is a unitary matrix that can be represented as:(1)$$U\left(\theta ,\phi ,\lambda \right)=\left(\begin{array}{cc} \cos(\frac{\theta }{2}) & -e^{i\lambda }\sin\left(\frac{\theta }{2}\right)\\ e^{i\phi }\sin\left(\frac{\theta }{2}\right) & e^{i(\phi +\lambda )}\cos\left(\frac{\theta }{2}\right) \end{array}\right)$$

We can already highlight the fact that while classic players are limited to only two pure strategies (e.g., cooperate or defect), quantum players can choose among an infinite number of pure strategies, that is, any combination of real value for the three parameters θ, ϕ and λ. The *fifth* step is to apply the operator $J^{†}$ (*J* conjugate transpose) after the strategies of the players. Finally, the *sixth* step consists of measuring the state of the qubits to read the classical outputs of the circuit and, therefore, the final action of each player. The readouts are used as inputs of the pay-off matrix to determine the players’ rewards.

One last thing to add is the fact that we are going to replace the three-parameter general one-qubit gate U(θ,ϕ,λ) with three one-parameter rotation one-qubit gates $R_{X}\left(\varphi_{1}\right)R_{Y}\left(\varphi_{2}\right)R_{X}\left(\varphi_{3}\right)$, with $R_{X}\left(\varphi \right)=exp\left(-i\frac{\varphi }{2}X\right)=\left(\begin{array}{cc} \cos(\frac{\varphi }{2}) & -i\sin(\frac{\varphi }{2})\\ -i\sin(\frac{\varphi }{2}) & \cos(\frac{\varphi }{2}) \end{array}\right)$ and $R_{Y}\left(\varphi \right)=exp\left(-i\frac{\varphi }{2}Y\right)=\left(\begin{array}{cc} \cos(\frac{\varphi }{2}) & -\sin(\frac{\varphi }{2})\\ \sin(\frac{\varphi }{2}) & \cos(\frac{\varphi }{2}) \end{array}\right)$. This is possible without losing generality since $U\left(\theta ,\phi ,\lambda \right)=e^{i\alpha }R_{\hat{n}}\left(\beta \right)R_{\hat{m}}\left(\gamma \right)R_{\hat{n}}\left(\delta \right)$ [36]. Having said that, the circuit from Figure 1 becomes the one from Figure 2.

### 2.2. Learning Algorithm

In this section, the algorithm employed by the agents for learning optimal actions in the quantum games previously presented is elucidated. The algorithm operates on a decentralized basis, with each player solely attempting to maximize their own long-term reward. Although players do not share any information with their opponents about the game they are engaged in or the rewards they receive, they have to share their actions with other participants.

Before delving into the algorithmic description, it is important to define a few crucial concepts. First, each player’s strategy is represented as the matrix product of three parameter-dependent (actions) quantum rotation gates: $u_{i}=R_{X}\left(\varphi_{i3}\right)\times R_{Y}\left(\varphi_{i2}\right)\times R_{X}\left(\varphi_{i1}\right)$. Second, the joint strategy of all players is represented as the Kronecker tensor product of all individual players’ actions. For the case of two players A and B, this is expressed as $U=u_{A}⊗u_{B}=\left(R_{X}\left(\varphi_{A3}\right)\times R_{Y}\left(\varphi_{A2}\right)\times R_{X}\left(\varphi_{A1}\right)\right)⊗\left(R_{X}\left(\varphi_{B3}\right)\times R_{Y}\left(\varphi_{B2}\right)\times R_{X}\left(\varphi_{B1}\right)\right)$. This formalism allows the encapsulation of the collective strategies of players in the quantum game.

The output quantum state of the quantum circuit is then computed as $|\psi_{out}\rangle =J^{†}\times U\times J\times |00\rangle$. Lastly, the probability of each possible classical state being measured at the end of the quantum circuit is determined by the Born rule $prob\left(\varphi_{A1},\varphi_{A2},\varphi_{A3},\varphi_{B1},\varphi_{B2},\varphi_{B3}\right)={\left|\psi_{out}\right|}^{2}=\left[\begin{array}{c} p_{00}\\ p_{01}\\ p_{10}\\ p_{11} \end{array}\right]$. This equation directly relates the actions of the players with the probabilistic outcomes of the quantum game. All these concepts are readily extensible to a general case involving N players.

All N-player games with two actions, as mentioned above, can be represented by a matrix of $2^{N}$ rows and *N* columns. Each row corresponds to a specific combination of actions chosen by the players, while each column represents the reward received by each player. This matrix representation serves as a succinct summary of all rewards assigned in the game for every possible combination of actions. As an illustrative example, a generic two-player game with two actions can be represented by the following matrix, where the pairs [Rai,Rbj] denote the reward combinations for player A and player B in response to their possible actions, respectively, $game=\left[\begin{array}{cc} Ra1 & Rb1\\ Ra2 & Rb2\\ Ra3 & Rb3\\ Ra4 & Rb4 \end{array}\right]$.

Using the previously defined concepts, a reward vector can be computed. This is achieved by multiplying the output state probability vector by the game matrix. The mathematical representation of this operation is given by $reward\left(\varphi_{A1},\varphi_{A2},\varphi_{A3},\varphi_{B1},\varphi_{B2},\varphi_{B3}\right)=prob.transpose\left(\right)*game$. In this calculation, the first row of the resulting vector represents the reward assigned to Player A: $reward_{a}=p_{00}*Ra1+p_{01}*Ra2+p_{10}*Ra3+p_{11}*Ra4$. This is a weighted sum of rewards for Player A, with weights corresponding to the probabilities of the respective states. Similarly, the second row corresponds to the reward for Player B: $reward_{b}=p_{00}*Rb1+p_{01}*Rb2+p_{10}*Rb3+p_{11}*Rb4$. Thus, the proposed framework allows for a clear determination of the expected rewards for each player given their actions.

It is now possible to outline the model-free policy-based algorithm employed by the players to update their strategies based on the rewards received in each iteration. Here, the term policy refers to a function used by the agent to select actions given the current reward received [37]. Every player starts with a randomly selected set of actions and utilizes the Adam algorithm [29], a prominent gradient-based optimization method in the machine learning domain, which adaptively adjusts the learning rates for the actions. However, before proceeding further, a critical aspect needs to be defined: the method by which each player calculates the gradient of the reward function with respect to its actions. The agent estimates the value of the gradient by recalculating its own reward but modifying one of their own actions by a small difference of ϵ. For instance, Player A, possessing information about the ongoing game and the actions taken by other players, can calculate the gradient with respect to each of their actions in the following manner:$$\nabla \varphi_{A1}=\frac{reward_{a}\left(\varphi_{A1}+\varepsilon ,\varphi_{A2},\varphi_{A3},\varphi_{B1},\varphi_{B2},\varphi_{B3}\right)-reward_{a}\left(\varphi_{A1},\varphi_{A2},\varphi_{A3},\varphi_{B1},\varphi_{B2},\varphi_{B3}\right)}{\varepsilon }$$
$$\nabla \varphi_{A2}=\frac{reward_{a}\left(\varphi_{A1},\varphi_{A2}+\varepsilon ,\varphi_{A3},\varphi_{B1},\varphi_{B2},\varphi_{B3}\right)-reward_{a}\left(\varphi_{A1},\varphi_{A2},\varphi_{A3},\varphi_{B1},\varphi_{B2},\varphi_{B3}\right)}{\varepsilon }$$
$$\nabla \varphi_{A3}=\frac{reward_{a}\left(\varphi_{A1},\varphi_{A2},\varphi_{A3}+\varepsilon ,\varphi_{B1},\varphi_{B2},\varphi_{B3}\right)-reward_{a}\left(\varphi_{A1},\varphi_{A2},\varphi_{A3},\varphi_{B1},\varphi_{B2},\varphi_{B3}\right)}{\varepsilon }$$

Upon calculation of the gradients, the players’ actions could be directly adjusted in the direction of increasing rewards using stochastic gradient ascent (SGD) straightforwardly as in (Equation 2), where α is the learning rate, that is, the speed at which a model adjusts to new feedback. Agents could also incorporate a moving average of the last gradients in order to take bigger steps on flatter areas and smaller steps in steeper areas, as in RMSProp.
(2)$$\left\{\begin{array}{c} \varphi_{A1}^{t+1}=\varphi_{A1}^{t}+\alpha *\nabla \varphi_{A1}\\ \varphi_{A2}^{t+1}=\varphi_{A2}^{t}+\alpha *\nabla \varphi_{A2}\\ \varphi_{A3}^{t+1}=\varphi_{A3}^{t}+\alpha *\nabla \varphi_{A3} \end{array}\right.\;$$

However, the Adam algorithm surpasses vanilla SGD and RMSProp by offering adaptive learning rates that adjust for each action individually to navigate the reward function, employing moving averages (as RMSProp), but also incorporates the calculation of moment estimation. Although it needs additional computations, which are defined next:$$\left\{\begin{array}{c} m_{A1}^{t+1}=\beta_{1}*m_{A1}^{t}+\left(1-\beta_{1}\right)*\nabla \varphi_{A1}\\ m_{A2}^{t+1}=\beta_{1}*m_{A2}^{t}+\left(1-\beta_{1}\right)*\nabla \varphi_{A2}\\ m_{A3}^{t+1}=\beta_{1}*m_{A3}^{t}+\left(1-\beta_{1}\right)*\nabla \varphi_{A3}\\ v_{A1}^{t+1}=\beta_{2}*v_{A1}^{t}+\left(1-\beta_{2}\right)*\nabla \varphi_{A1}^{2}\\ v_{A2}^{t+1}=\beta_{2}*v_{A2}^{t}+\left(1-\beta_{2}\right)*\nabla \varphi_{A2}^{2}\\ v_{A3}^{t+1}=\beta_{2}*v_{A3}^{t}+\left(1-\beta_{2}\right)*\nabla \varphi_{A3}^{2} \end{array}\right.$$
$$\left\{\begin{array}{c} {\hat{m}}_{A1}=m_{A1}^{t+1}/\left(1-\beta_{1}^{t+1}\right)\\ {\hat{m}}_{A2}=m_{A2}^{t+1}/\left(1-\beta_{1}^{t+1}\right)\\ {\hat{m}}_{A3}=m_{A3}^{t+1}/\left(1-\beta_{1}^{t+1}\right)\\ {\hat{v}}_{A1}=v_{A1}^{t+1}/\left(1-\beta_{2}^{t+1}\right)\\ {\hat{v}}_{A2}=v_{A2}^{t+1}/\left(1-\beta_{2}^{t+1}\right)\\ {\hat{v}}_{A3}=v_{A3}^{t+1}/\left(1-\beta_{2}^{t+1}\right) \end{array}\right.$$
$$\left\{\begin{array}{c} \varphi_{A1}^{t+1}=\varphi_{A1}^{t}+\left(\alpha_{1}*{\hat{m}}_{A1}\right)/\left(\sqrt{{\hat{v}}_{A1}}+\alpha_{2}\right)\\ \varphi_{A2}^{t+1}=\varphi_{A2}^{t}+\left(\alpha_{1}*{\hat{m}}_{A2}\right)/\left(\sqrt{{\hat{v}}_{A2}}+\alpha_{2}\right)\\ \varphi_{A3}^{t+1}=\varphi_{A3}^{t}+\left(\alpha_{1}*{\hat{m}}_{A3}\right)/\left(\sqrt{{\hat{v}}_{A3}}+\alpha_{2}\right) \end{array}\right.$$
where $\alpha_{1}$ determines the step size at each iteration. $\alpha_{2}$ is a very small number to prevent any division by zero. $\beta_{1}$ is used for the exponentially decaying average of past gradients. $\beta_{2}$ is used for the exponentially decaying average of past squared gradients. Finally, *m* is an estimate of the first moment (the mean) of the gradients and *v* is an estimate of the second moment (the uncentered variance) of the gradients. For further information on the Adam algorithm, please refer to the bibliography [29].

The gradient calculation method and the Adam algorithm both involve various hyperparameters. These values have been meticulously studied, resulting in the identification of an optimal set of values. These optimal values are adopted for the remainder of the article, ensuring consistency and optimal results in subsequent analyses and discussions.
$$\left\{\begin{array}{c} \varepsilon =1\times 10^{8}\\ \alpha_{1}=0.0001\\ \alpha_{2}=1\times 10^{8}\\ \beta_{1}=0.9\\ \beta_{2}=0.999 \end{array}\right.$$

Finally, a block diagram depicting the comprehensive system that includes both the learning agents and the quantum game environment can be found in Figure 3.

## 3. Results

This section presents the results obtained from applying the previously described algorithm to the general-sum games mentioned earlier. Graphical representations of the rewards for each player over 1,000,000 iterations are provided for two scenarios:Games without entanglement, achieved with a value of γ=0 in the operator *J*. This scenario corresponds to the classical version of the game.Games with entanglement, achieved with a value of $\gamma =\frac{\pi }{2}$ in the operator *J*, which corresponds to the fully quantum version of the game.

The outcomes are presented in this way to verify the algorithm’s proper functionality—the classical scenario should behave as expected—and to compare the dynamics and performance of the players between the classical and quantum cases. This comparison can reveal any noteworthy shifts in behavior or strategy that emerge in the transition from a classical to a quantum framework.

This analysis begins with the minority game. Figure 4 illustrates the evolution of the player rewards for both the classical (left) and quantum (right) cases, specifically for games that involve three, four and five players. Note that the minority game does not have a meaningful interpretation for a two-player scenario.

The results are indeed compelling, although a more specific description of the minority game is beneficial for our analysis. As framed in our environment, the minority game for N players provides each player with two options (0 and 1), while a reward of $R_{N}=10$ is conferred to the players who choose the minority option, which is defined as any selection made by fewer than 50% of the players [32].

Figure 4 shows that the average rewards for the game involving three and five players are nearly identical in both the classical and quantum cases ($R_{3}=\frac{10*1}{3}=3.333$ and $R_{5}=\frac{10*2}{5}=3.999$ versus $R_{3}=3.333$ and $R_{5}=3.999$, respectively), exactly as the classical and quantum Nash equilibria predicted in [3]. Not only are the values almost the same, but they also both converge to an optimal equilibrium state: in a three-player game, a single player is in the minority, while in a five-player game, two players constitute the minority. This optimal equilibrium denotes a state of balance where rewards are maximized, underscoring the algorithm’s successful implementation in both classical and quantum contexts.

However, for the four-player game, the classical case converges to a reward of $R_{3}\approx 0$ for all players, while the quantum case does not. Players in the quantum game manage to learn a set of strategies, ensuring that each player wins with a probability p≈1/4, also as predicted in [3]. This results in each player receiving an average reward of roughly $R_{4}=\frac{10*1}{4}\approx 2.2451$, demonstrating a clear advantage of quantum gameplay over the classical approach in this particular scenario.

Let us now turn our attention to Figure 5, which depicts the evolution of player rewards in the Platonia game for both classical (left) and quantum (right) setups, for two, three, four and five players. In this game, a reward of $R_{N}=10$ is given only to a player who selects action 1 but only if all other players have chosen option 0, as depicted in Table 8.

The results of this game are certainly more striking. All players in the classical game converge (regardless of the number of players, N) to their Nash equilibrium, resulting in a total reward of $R_{N}\approx 0$ for all players. Despite being predictable, this outcome resulted in the worst possible performance. In stark contrast, the quantum setup showcases the power of entanglement, as players consistently achieve significantly higher total reward, irrespective of the number of players involved ($R_{2}=9.66$, $R_{3}=9.99$, $R_{4}=4.21$ and $R_{5}=9.58$). This remarkable result highlights the effectiveness of the algorithm and the potential of quantum strategies in complex multiplayer games, which may offer advantages over their classical counterparts.

Figure 6 depicts the evolution of rewards for players in both the classical (left) and quantum (right) scenarios of the unscrupulous diners dilemma game. This game involves two, three, four and five players and it is an N-player extension of the well-known prisoners’ dilemma. The unscrupulous diners dilemma represents a situation where a group, agreeing to split the cost of a meal, must individually choose between a less expensive or a more expensive dish. While choosing the more expensive dish may seem advantageous to each individual, this collective decision ultimately leads to a less optimal overall financial outcome [34]. In Table 9, we present a payoff matrix for the two-player version of the unscrupulous diners game, which also serves as a representation of the renowned prisoners’ dilemma.

As seen in Figure 6, the rewards of the classical players always (for every *N*) quickly converge to the value of $R_{N}\approx 3.33$. This value represents the expected classical Nash equilibrium and corresponds to the selfish action that yields the minimum reward when chosen by all players. However, the quantum case presents another encouraging contrast. In the case of two players, both rewards converge to a value of $R_{2}\approx 6.66$, corresponding to the ideal value associated with both players cooperating. On the other hand, for three, four and five players, the rewards of the quantum players do not completely converge; instead, they fluctuate around achieving the final values of $R_{N}=\left[R_{3},R_{4},R_{5}\right]=\left[5.55,5.07,5.32\right]$. Although this value is not the highest possible, it is significantly greater than the $R_{N}\approx 3.33$ obtained in the classical case.

Finally, we focus on the volunteer game, as depicted in Figure 7. The volunteer game is an N-player extension of the chicken game, embodying scenarios where an entire group benefits from an individual’s minor sacrifice, but all suffer if no one acts [35]. The payoff matrix for the two-player volunteer dilemma game is illustrated in Table 10, representing at the same time the famous chicken game.

In this final scenario, we observe that the behaviors of both the classical and quantum cases are nearly identical for every N. Rewards consistently converge to the expected value of the classical Nash equilibrium $R_{N}=1/N$ in both setups. This outcome thereby confirms that quantum advantages may not always be present and that our proposed algorithm is a valuable tool for efficiently and systematically verifying these behaviors.

All the computer code required to reproduce the results reported in this section can be found in the Github repository shared in the “Appendix A” section.

## 4. Noise Effects

This chapter examines how the inherent noise in quantum computers might affect the performance of the learning algorithm. This analysis is a first step towards understanding the practical implications of deploying such algorithms in real-world quantum computational environments, where noise is an unavoidable factor.

In Section 3, we assumed that, after the application of operator *J*, all players could apply their gates to their qubits in an ideal channel, without any type of noise affecting the vulnerability of the quantum system. Given the difficulty of meeting such a condition, it becomes necessary to undertake a modeling approach to take into account this situation accurately.

The noise model chosen for our study is the depolarizing channel, primarily because it effectively models both bit-flip and phase-flip errors. While the depolarizing channel captures the essential features of numerous real-world noise processes, it is a simplified model. In order to represent all the nuances of noise in some specific quantum devices, further fine-tuning is needed. The state of the quantum system of one qubit after this noise is $\varepsilon \left(\rho \right)=\left(1-\lambda \right)\rho +\frac{\lambda I}{2}=\left(1-\lambda \right)\rho +\frac{\lambda }{3}\left(X\rho X+Y\rho Y+Z\rho Z\right)$, with ρ=|ψ〉〈ψ| being the density matrix of the quantum state before the noise is applied. The way to model this (Figure 8) is by adding a fourth gate after $R_{X}\left(\varphi_{1}\right)R_{Y}\left(\varphi_{2}\right)R_{X}\left(\varphi_{3}\right)$; this gate will be selected randomly following the probabilities: 

$U4=\left\{\begin{array}{cc} I=\left[\begin{array}{cc} 1 & 0\\ 0 & 1 \end{array}\right] & \;\mathrm{with}\;p=(1-\lambda )\\ X=\left[\begin{array}{cc} 0 & 1\\ 1 & 0 \end{array}\right] & \;\mathrm{with}\;p=\frac{\lambda }{3}\\ Y=\left[\begin{array}{cc} 0 & -i\\ i & 0 \end{array}\right] & \;\mathrm{with}\;p=\frac{\lambda }{3}\\ Z=\left[\begin{array}{cc} 1 & 0\\ 0 & -1 \end{array}\right] & \;\mathrm{with}\;p=\frac{\lambda }{3} \end{array}\right.$.

**Figure 8 entropy-25-01484-f008:**
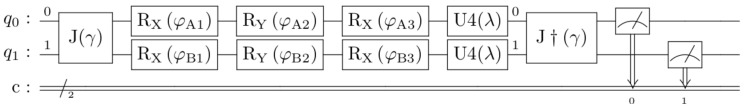
A comprehensive model of the EWL protocol for two players, incorporating both the entanglement factor and the effects of depolarizing channel noise.

This means that the quantum state in each channel will remain intact with probability (1−λ) and will be modified with probability λ. To modify the quantum state of the channel, the X, Y or Z gates will be applied with the same probability.

For this analysis, our focus will be on the Platonia game in its quantum variant ($\gamma =\frac{\pi }{2}$), as this game manifests the most significant advantage when contrasting classical and quantum models. Figure 9 and Figure 10 illustrate the variation in the average reward of players as a function of the quantum noise (λ and log(λ), respectively, and the red dotted line represents the noiseless system, λ=0) inherent in the quantum circuit of the EWL model for two, three, four, five, six and seven players.

On the one hand, for the scenarios involving two, three, four and five players, an expected outcome is observed: as the noise in the quantum circuit increases, the performance of the players deteriorates. This is anticipated since the feedback received from the players to update their strategies, using the proposed algorithm, contains progressively more noise as λ increases. Consequently, the learning efficacy of the players decreases with increasing λ, leading to a decrease in their overall performance. All players achieve a maximum average reward when λ=0, with the ideal case of no noise.

On the other hand, a particular behavior is observed when the number of agents in the quantum game rises to six and seven players. It appears that a slight degree of noise in the quantum circuit is advantageous for the algorithm in terms of maximizing the players’ average reward. In both instances, games involving six and seven players yield the highest average reward for a non-zero value of λ (λ=0.0078125 and λ=0.00390625, respectively).

It is important to note that there could be a trivial case in which the performance of the players improves in a non-decreasing manner as the amount of noise in the quantum circuit increases. That is, in several games, a player acting entirely at random (maximum noise) achieves a higher reward than a rational player (zero noise). However, such behavior would not be particularly noteworthy. For example, this is observed in the classic prisoners’ dilemma (Table 9), where rational behavior converges on a reward of R=3.33, while completely random behavior yields a reward of $R=\frac{0+3.33+6.66+10}{4}=4$. In our study, a small amount of noise is beneficial, but excessive noise remains disadvantageous. This nontrivial behavior is relevant since any non-isolated quantum system will have intrinsic noise that could potentially be harnessed. This is particularly noteworthy in the present context, where all contemporary quantum computers inevitably possess a significant level of noise. Our quantum noise model is very general and does not represent any particular type of quantum device made of different material. Therefore, discovering applications where such noise can be leveraged to enhance system performance is of utmost relevance.

A plausible explanation for this phenomenon might stem from the fact that when the number of players increases, the reward function becomes more complex and, if the algorithm operates in a noiseless setup, it can become trapped in local maxima. However, a minuscule amount of noise in the quantum circuits might suffice to remove the system from these local maxima, allowing exploration of other actions with higher rewards. Importantly, the noise level needs to be low enough to not harm the overall learning process and therefore the performance of the system.

## 5. Conclusions

This research has probed the intricate dynamics of quantum games involving multiple agents, shedding light on the complex relationships between quantum noise, system performance and learning efficacy. It has been shown that gradient-based strategies in a multi-agent setting present intriguing characteristics when quantum mechanics are taken into account.

The proposed quantum learning algorithm has shown robust performance across various game configurations. Furthermore, the investigation of quantum noise effects uncovered a unique behavior that might have far-reaching implications. As expected, increased noise caused player performance in games involving fewer participants to deteriorate. However, strikingly, a slight amount of noise appeared advantageous in games with six and seven players, seemingly facilitating escape from local maxima and enabling exploration of more rewarding actions. This discovery stimulates discussion on the potential leverage of system noise, a common challenge in contemporary quantum computers, to enhance performance.

The insights from this study hold substantial implications for the real-world application of quantum computing. For example, in economic and financial models where strategic interaction and learning play critical roles, the exploitation of quantum effects could lead to more realistic and accurate predictions. In machine learning and artificial intelligence, quantum enhancements might allow for more efficient solution exploration and optimization. Finally, in communication networks, the principles of quantum games could enable the development of more robust quantum-enhanced communication systems.

The exploration of quantum games with multi-agent learning in this research has only scratched the surface. There is a wealth of complex dynamics and fascinating phenomena to be explored in this burgeoning field. Further research is anticipated to elucidate these exciting frontiers, advancing our understanding of quantum games and their potential applications.

## Figures and Tables

**Figure 1 entropy-25-01484-f001:**
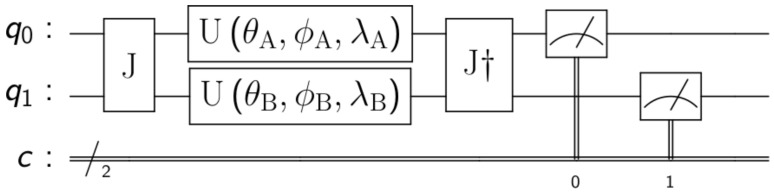
EWL game model for 2 players. Where $q_{0}$ and $q_{1}$ are the initial quantum states of the players and *c* is a classical register where the qubit measurements are stored.

**Figure 2 entropy-25-01484-f002:**
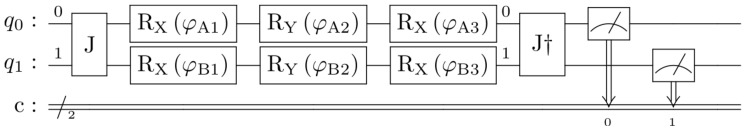
EWL game model for 2 players with rotation gates.

**Figure 3 entropy-25-01484-f003:**
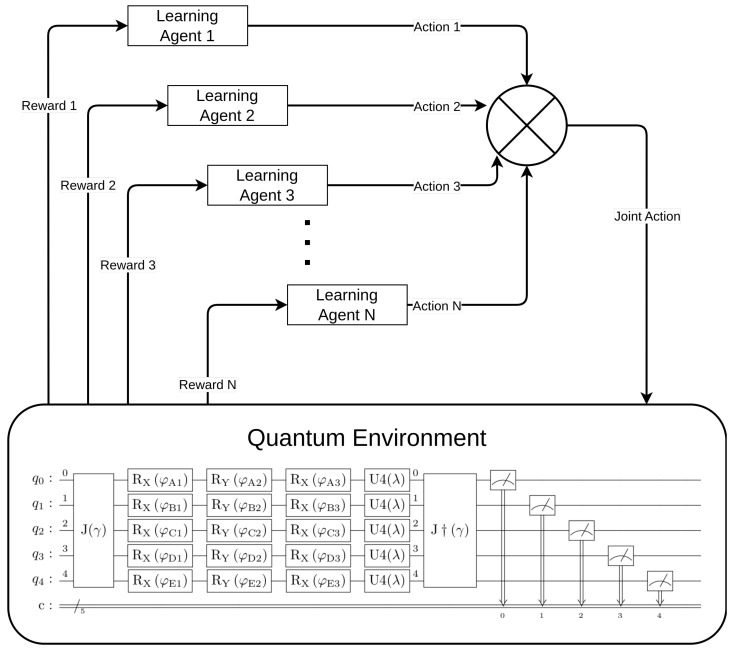
Complete model of the system: agents learning + quantum environment.

**Figure 4 entropy-25-01484-f004:**
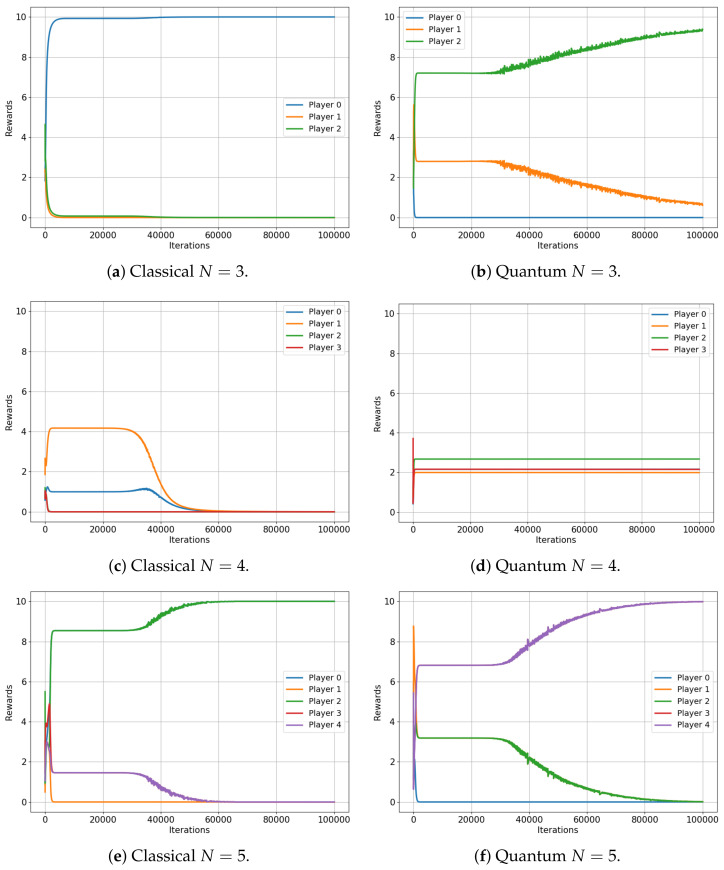
Classical versus Quantum learning of Minority Games for 3, 4 and 5 players. (**a**): Average reward =3.3333. (**b**): Average reward =3.3333. (**c**): Average reward =0.0007. (**d**): Average reward =2.2451. (**e**): Average reward =3.9999. (**f**): Average reward =3.9999.

**Figure 5 entropy-25-01484-f005:**
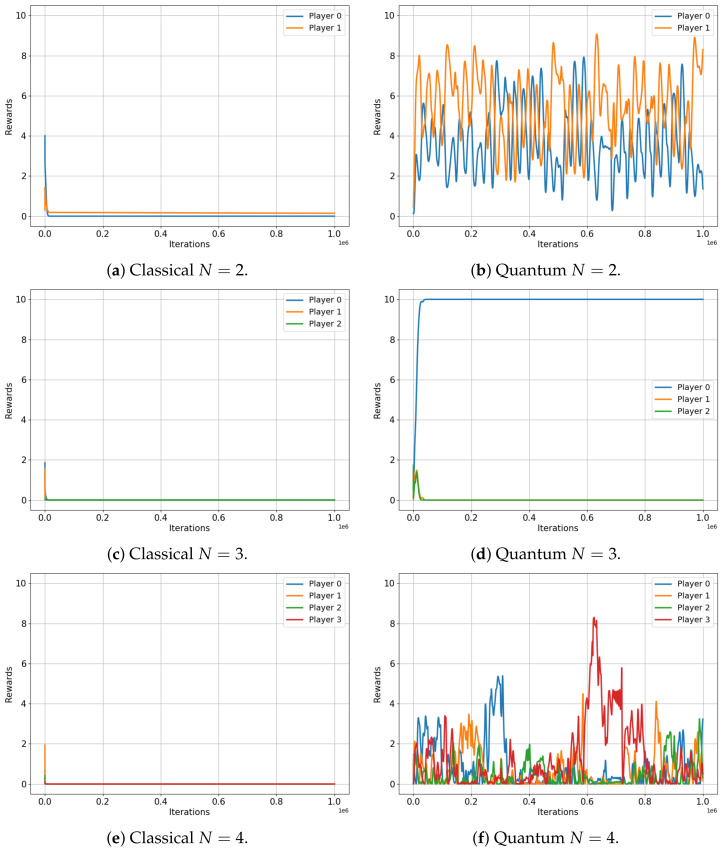
Classical versus Quantum learning of Platonia Games for 2, 3, 4 and 5 players. (**a**) Total reward =0.1445. (**b**) Total reward =9.6604. (**c**) Total reward =0.0090. (**d**) Total reward =9.9999. (**e**) Total reward =0.0008. (**f**) Total reward =4.2078. (**g**) Total reward =0.0001. (**h**) Total reward =9.5895.

**Figure 6 entropy-25-01484-f006:**
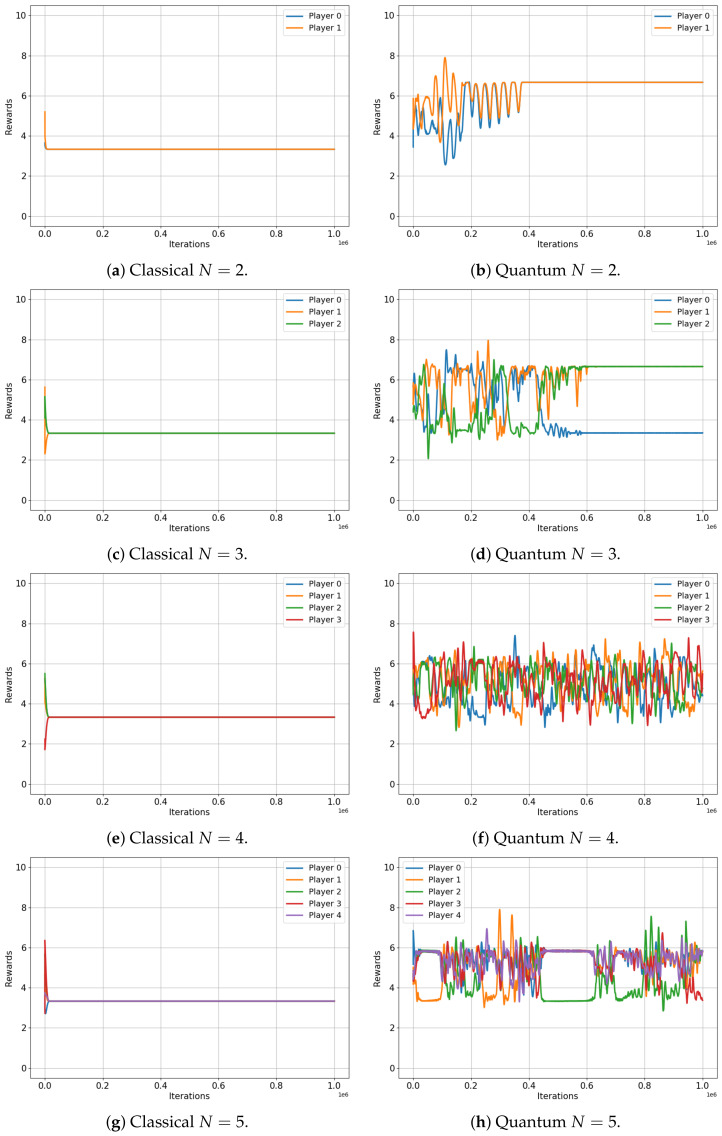
Classical versus Quantum learning of Unscrupulous Games for 2, 3, 4 and 5 players. (**a**) Average reward =3.3333. (**b**) Average reward =6.6666. (**c**) Average reward =3.3333. (**d**) Average reward =5.5525. (**e**) Average reward =3.3333. (**f**) Average reward =5.0717. (**g**) Average reward =3.3333. (**h**) Average reward =5.3244.

**Figure 7 entropy-25-01484-f007:**
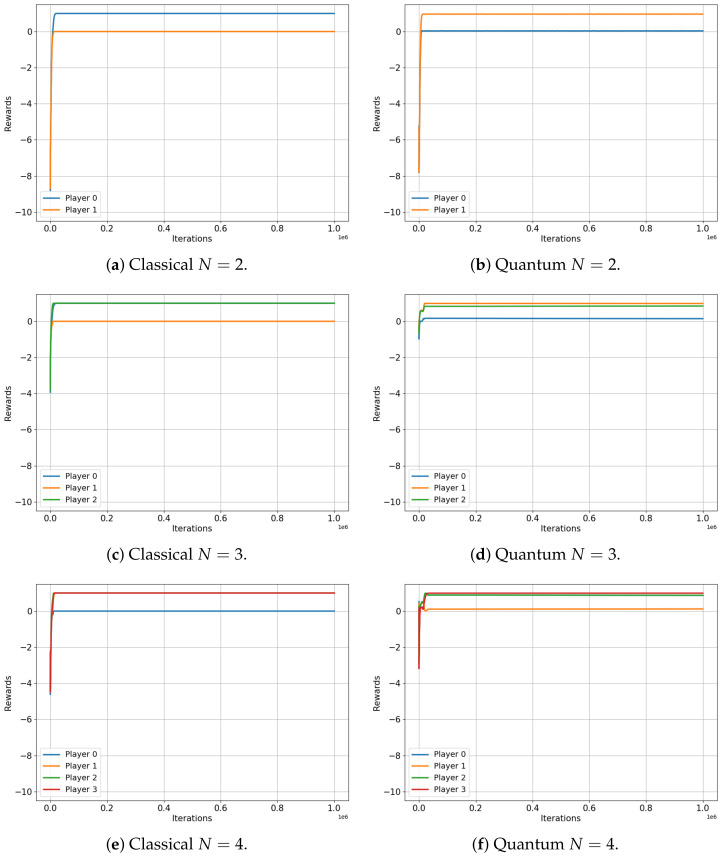
Classical versus Classical learning of Volunteer Games for 2, 3, 4 and 5 players. (**a**) Average reward =0.4999. (**b**) Average reward =0.4999. (**c**) Average reward =0.6666. (**d**) Average reward =0.6606. (**e**) Average reward =0.7499. (**f**) Average reward =0.7164. (**g**) Average reward =0.7999. (**h**) Average reward =0.7390.

**Figure 9 entropy-25-01484-f009:**
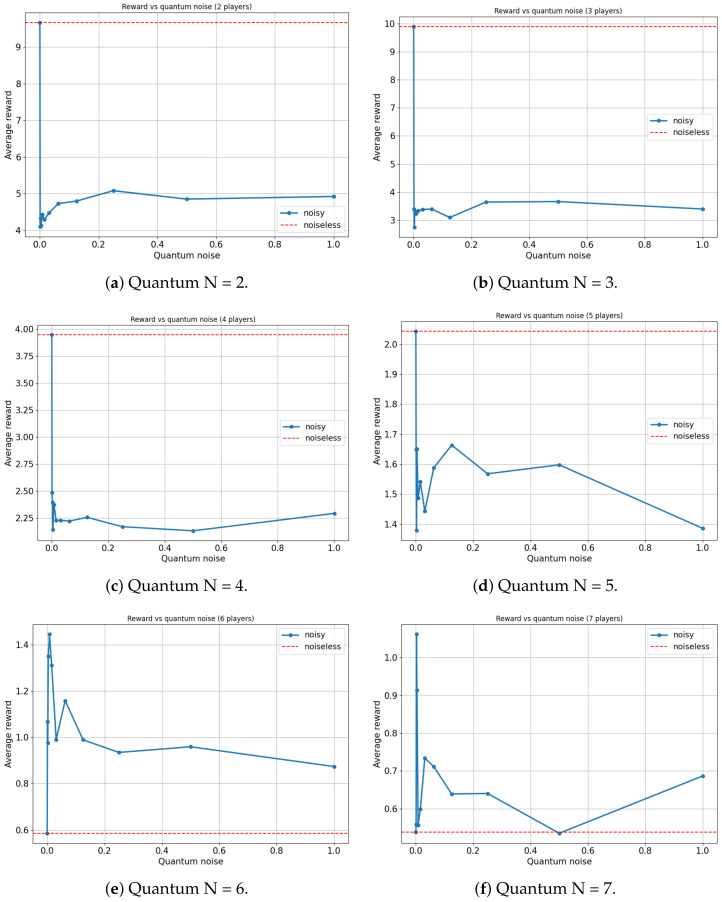
Average reward of N-player Platonia Game versus Quantum noise plotted as λ (for $\lambda =[0,\frac{1}{1024},\frac{1}{512},\frac{1}{256},\frac{1}{128},\frac{1}{64},\frac{1}{32},\frac{1}{16},\frac{1}{8},\frac{1}{4},\frac{1}{2},1]$). (**a**) Maximum average reward = 9.882 for quantum noise for λ = 0. (**b**) Maximum average reward = 9.971 for quantum noise for λ = 0. (**c**) Maximum average reward = 4.319 for quantum noise for λ = 0. (**d**) Maximum average reward = 6.451 for quantum noise for λ = 0. (**e**) Maximum average reward = 2.221 for quantum noise for λ = 0.0078125. (**f**) Maximum average reward = 1.686 for quantum noise for λ = 0.00390625.

**Figure 10 entropy-25-01484-f010:**
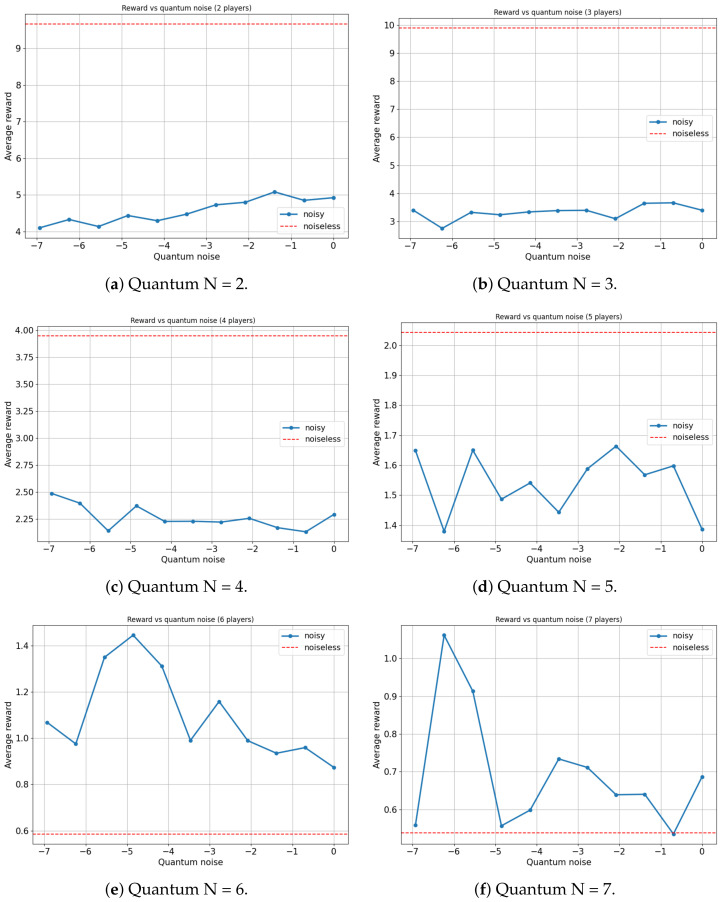
Average reward of N-player Platonia Game versus Quantum noise plotted as log(λ) (for $\lambda =[0,\frac{1}{1024},\frac{1}{512},\frac{1}{256},\frac{1}{128},\frac{1}{64},\frac{1}{32},\frac{1}{16},\frac{1}{8},\frac{1}{4},\frac{1}{2},1]$). (**a**) Maximum average reward = 9.882 for quantum noise for λ = 0. (**b**) Maximum average reward = 9.971 for quantum noise for λ = 0. (**c**) Maximum average reward = 4.319 for quantum noise for λ = 0. (**d**) Maximum average reward = 6.451 for quantum noise for λ = 0. (**e**) Maximum average reward = 2.221 for quantum noise for λ = 0.0078125. (**f**) Maximum average reward = 1.686 for quantum noise for λ = 0.00390625.

**Table 1 entropy-25-01484-t001:** General payoff matrix for 2 players and 2 strategies.

Strategies	Rewards
(A0, B0)	(Ra1, Rb1)
(A0, B1)	(Ra2, Rb2)
(A1, B0)	(Ra3, Rb3)
(A1, B1)	(Ra4, Rb4)

**Table 2 entropy-25-01484-t002:** General payoff matrix for 3 players and 2 strategies.

Strategies	Rewards
(A0, B0, C0)	(Ra1, Rb1, Rc1)
(A0, B0, C1)	(Ra2, Rb2, Rc2)
(A0, B1, C0)	(Ra3, Rb3, Rc3)
(A0, B1, C1)	(Ra4, Rb4, Rc4)
(A1, B0, C0)	(Ra5, Rb5, Rc5)
(A1, B0, C1)	(Ra6, Rb6, Rc6)
(A1, B1, C0)	(Ra7, Rb7, Rc7)
(A1, B1, C1)	(Ra8, Rb8, Rc8)

**Table 3 entropy-25-01484-t003:** General payoff matrix for 4 players and 2 strategies.

Strategies	Rewards
(A0, B0, C0, D0)	(Ra1, Rb1, Rc1, Rd1)
(A0, B0, C0, D1)	(Ra2, Rb2, Rc2, Rd2)
(A0, B0, C1, D0)	(Ra3, Rb3, Rc3, Rd3)
(A0, B0, C1, D1)	(Ra4, Rb4, Rc4, Rd4)
(A0, B1, C0, D0)	(Ra5, Rb5, Rc5, Rd5)
(A0, B1, C0, D1)	(Ra6, Rb6, Rc6, Rd6)
(A0, B1, C1, D0)	(Ra7, Rb7, Rc7, Rd7)
(A0, B1, C1, D1)	(Ra8, Rb8, Rc8, Rd8)
(A1, B0, C0, D0)	(Ra9, Rb9, Rc9, Rd9)
(A1, B0, C0, D1)	(Ra10, Rb10, Rc10, Rd10)
(A1, B0, C1, D0)	(Ra11, Rb11, Rc11, Rd11)
(A1, B0, C1, D1)	(Ra12, Rb12, Rc12, Rd12)
(A1, B1, C0, D0)	(Ra13, Rb13, Rc13, Rd13)
(A1, B1, C0, D1)	(Ra14, Rb14, Rc14, Rd14)
(A1, B1, C1, D0)	(Ra15, Rb15, Rc15, Rd15)
(A1, B1, C1, D1)	(Ra16, Rb16, Rc16, Rd16)

**Table 4 entropy-25-01484-t004:** Minority game payoff matrix for 3 players.

Strategies	Rewards
(0, 0, 0)	(0, 0, 0)
(0, 0, 1)	(0, 0, 10)
(0, 1, 0)	(0, 10, 0)
(0, 1, 1)	(10)
(1, 0, 0)	(10, 0, 0)
(1, 0, 1)	(0, 10, 0)
(1, 1, 0)	(0, 0, 10)
(1, 1, 1)	(0, 0, 0)

**Table 5 entropy-25-01484-t005:** Platonia dilemma payoff matrix for 3 players.

Strategies	Rewards
(0, 0, 0)	(0, 0, 0)
(0, 0, 1)	(0, 0, 10)
(0, 1, 0)	(0, 10, 0)
(0, 1, 1)	(0, 0, 0)
(1, 0, 0)	(10, 0, 0)
(1, 0, 1)	(0, 0, 0)
(1, 1, 0)	(0, 0, 0)
(1, 1, 1)	(0, 0, 0)

**Table 6 entropy-25-01484-t006:** Unscrupulous dilemma payoff matrix for 3 players.

Strategies	Rewards
(0, 0, 0)	(6.66, 6.66, 6.66)
(0, 0, 1)	(3.33, 3.33, 10.0)
(0, 1, 0)	(3.33, 10.0, 3.33)
(0, 1, 1)	(0.00, 6.66, 6.66)
(1, 0, 0)	(10.0, 3.33, 3.33)
(1, 0, 1)	(6.66, 0.00, 6.66)
(1, 1, 0)	(6.66, 6.66, 0.00)
(1, 1, 1)	(3.33, 3.33, 3.33)

**Table 7 entropy-25-01484-t007:** Volunteers’ dilemma payoff matrix for 3 players.

Strategies	Rewards
(0, 0, 0)	(0, 0, 0)
(0, 0, 1)	(0, 0, 1)
(0, 1, 0)	(0, 1, 0)
(0, 1, 1)	(0, 1, 1)
(1, 0, 0)	(1, 0, 0)
(1, 0, 1)	(1, 0, 1)
(1, 1, 0)	(1, 1, 0)
(1, 1, 1)	(−10, −10, −10)

**Table 8 entropy-25-01484-t008:** Payoff matrix for N-player Platonia Game.

Player i\Other (N-1) Players	All 0	Some 1
0	0	0
1	10	0

**Table 9 entropy-25-01484-t009:** Payoff matrix for 2-player Unscrupulous Game or Prisoners Dilemma.

A\B	0	1
0	(6.66; 6.66)	(0.00; 10.0)
1	(10.0; 0.00)	(3.33; 3.33)

**Table 10 entropy-25-01484-t010:** Payoff matrix for 2-player Volunteer Dilemma or Chicken Game.

A\B	0	1
0	(0; 0)	(−1, 1)
1	(1; −1)	(−10; −10)

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
