# Peer review of "Maximizing Local Rewards on Multi-Agent Quantum Games through Gradient-Based Learning Strategies"

_entropy, 2023, doi:10.3390/e25111484_

Round 1

Reviewer 1 Report

This article explores the intricate domain of quantum games within multi-agent settings. The authors propose a novel model in which agents entangled via quantum circuits employ gradient-based strategies to optimize local rewards. The paper introduces a learning model to assess agents' learning efficacy across different games, focusing on the influence of quantum circuit noise on algorithm performance.

One of the primary findings of this research is the revelation of a non-trivial relationship between quantum circuit noise and algorithm performance. Although it is generally expected that increased quantum noise leads to a decline in performance, the authors present compelling evidence that low noise can unexpectedly enhance performance, particularly in games involving a substantial number of agents under specific conditions.

One minor thing that I believe the authors can consider is whether we can use optimizers other than the Adam to optimize the parameters? For example, the RMSProp or SGD? Or even gradient-free methods? Probably these aspects can be discussed further.

Author Response

Author's Reply to the Review Report (Reviewer 1):

This article explores the intricate domain of quantum games within multi-agent settings. The authors propose a novel model in which agents entangled via quantum circuits employ gradient-based strategies to optimize local rewards. The paper introduces a learning model to assess agents' learning efficacy across different games, focusing on the influence of quantum circuit noise on algorithm performance.

One of the primary findings of this research is the revelation of a non-trivial relationship between quantum circuit noise and algorithm performance. Although it is generally expected that increased quantum noise leads to a decline in performance, the authors present compelling evidence that low noise can unexpectedly enhance performance, particularly in games involving a substantial number of agents under specific conditions.

  • We sincerely appreciate your thoughtful review and the positive feedback you have provided on our paper. Your insights are very valuable for improving the quality of our work, and we are grateful for your time and effort in evaluating our research.

One minor thing that I believe the authors can consider is whether we can use optimizers other than the Adam to optimize the parameters? For example, the RMSProp or SGD? Or even gradient-free methods? Probably these aspects can be discussed further.

  • Thank you for your suggestion regarding the choice of optimizers in our paper. We appreciate your input and agree that discussing alternative optimizers could add value to our work. We added a discussion to justify the reasons why we decided to use Adam over SGD and RMSProp (from line 221 to 229) and mentioned other gradient-free methods in the literature previous to our work (lines 64 to 89).

Reviewer 2 Report

The authors attempt to build a quantum game theory following the quantum reinforcement learning multiple learning agents in this work. In particular, this work demonstrates that small quantum noises can even benefit the performance of quantum game theory based on quantum reinforcement learning, highlighting the significance of the proposed work on NISQ devices.

Strengths:

1. The descriptions of classical and quantum game theory are well written, where the Nash equilibrium in the quantum context is naturally introduced. 

2. It is an interesting finding that quantum noise can somehow benefit the empirical performance improvement of quantum reinforcement learning. 

3. The experimental simulations are comprehensive to verify their theoretical claim. 

Weaknesses:

1. It is still unknown where the benefits of quantum noise come from. It is suggested that the authors refer to the following two recent theory works on quantum machine learning to explain the 

Ref. 1: Caro, M.C., Huang, H.Y., Cerezo, M., Sharma, K., Sornborger, A., Cincio, L. and Coles, P.J., 2022. Generalization in quantum machine learning from few training data. Nature communications13(1), p.4919.

Ref. 2: Qi, J., Yang, C.H.H., Chen, P.Y. and Hsieh, M.H., 2023. Theoretical error performance analysis for variational quantum circuit based functional regression. npj Quantum Information9(1), p.4.

2. Besides, quantum devices made of different materials (superconducting, Ion-Trap, and Quantum Dot) own different quantum noise with unique characteristics, it is also unknown that the theoretical claim in this work is universal to the quantum noise of various quantum computers. 

Author Response

Author's Reply to the Review Report (Reviewer 2):

The authors attempt to build a quantum game theory following the quantum reinforcement learning multiple learning agents in this work. In particular, this work demonstrates that small quantum noises can even benefit the performance of quantum game theory based on quantum reinforcement learning, highlighting the significance of the proposed work on NISQ devices.

Strengths:

  1. The descriptions of classical and quantum game theory are well written, where the Nash equilibrium in the quantum context is naturally introduced. 
  2. It is an interesting finding that quantum noise can somehow benefit the empirical performance improvement of quantum reinforcement learning. 
  3. The experimental simulations are comprehensive to verify their theoretical claim. 

  • Thank you for your review of our paper. We appreciate your positive feedback on the clarity of our descriptions of classical and quantum game theory, as well as our findings regarding the potential benefit of quantum noise in quantum reinforcement learning. We are committed to addressing your suggestions in our revised manuscript to improve the paper further.

Weaknesses:

  1. It is still unknown where the benefits of quantum noise come from. It is suggested that the authors refer to the following two recent theory works on quantum machine learning to explain the:

Ref. 1: Caro, M.C., Huang, H.Y., Cerezo, M., Sharma, K., Sornborger, A., Cincio, L. and Coles, P.J., 2022. Generalization in quantum machine learning from few training data. Nature communications, 13(1), p.4919.

Ref. 2: Qi, J., Yang, C.H.H., Chen, P.Y. and Hsieh, M.H., 2023. Theoretical error performance analysis for variational quantum circuit based functional regression. npj Quantum Information, 9(1), p.4.

  1. Besides, quantum devices made of different materials (superconducting, Ion-Trap, and Quantum Dot) own different quantum noise with unique characteristics, it is also unknown that the theoretical claim in this work is universal to the quantum noise of various quantum computers.
  • Thank you for your insightful feedback on our paper. We greatly appreciate your suggestions, and we want to confirm that we have already incorporated the references you recommended ([16] and [23]) to provide a more comprehensive explanation of the benefits of quantum machine learning. Furthermore, we have incorporated a more complete discussion about the implications and limitations of our depolarizing noise model in the context of different quantum devices to address the variability of quantum noise across different platforms (Lines 330 to 334 and lines 368 to 374).

Reviewer 3 Report

The main of the present paper is to study the effects of noise on the learning performance of agent in the case of strategy for quantum games. The paper has some interesting observations. In my opinion, that paper should be improved  and I suggest to focus on the following aspects.

1. The Authors clam that there are some 'practical implications' applications of the presented results. However, I cannot see how this is the case. Noise is indeed present in NISQ systems, but it is also present in any non-isolated system. More in-depth explanation in this regard would be required.

2. Results are presented very poorly. Fonts in the figures are too small to read, the lack of grid makes the values hard to read on the plots, and there is almost no explanation of the results and their relevance.  This aspect should be improved before the resubmission.

3. There are some other attempts to use multi-agent approach in quantum games.
For example:

- J.A. Miszczak, L. Pawela, J. Sladkowski, General model for a
  entanglement-enhanced composed quantum game on a two-dimensional lattice,
  Fluctuation and Noise Letters, Vol. 13, No. 2 (2014), pp. 1450012

- R. Alonso-Sanz. A cellular automaton implementation of a quantum battle of the
  sexes game with imperfect information, Quantum Information Processing, Vol.
  14, pp. 3639–3659 (2015)

- L. Pawela, Quantum games on evolving random networks, Physica A: Statistical
  Mechanics and its Applications, Vol. 458, pp. 179-188 (2016)

What is (if any) relation of the presented research with the mentioned
references? Comments in this regard would be required.

4. No source code for the perforce numerical experiments is provided. This makes the research unrepeatable, and hence not very useful for further investigation. The paper cannot be published without providing source code and scripts used to obtain the presented results.

Author Response

Author's Reply to the Review Report (Reviewer 3):

The main of the present paper is to study the effects of noise on the learning performance of agent in the case of strategy for quantum games. The paper has some interesting observations. In my opinion, that paper should be improved  and I suggest to focus on the following aspects.

  • Thank you for your valuable feedback on our paper. We appreciate your comments, which have guided our revisions.

  1. The Authors clam that there are some 'practical implications' applications of the presented results. However, I cannot see how this is the case. Noise is indeed present in NISQ systems, but it is also present in any non-isolated system. More in-depth explanation in this regard would be required.

  • We have revised the paper to provide a more detailed explanation of the practical implications of our results in the context of quantum noise, addressing your concern about its relevance beyond NISQ systems. We have incorporated a more complete discussion about the implications and limitations of our depolarizing noise model in the context of different non-isolated systems to address the variability of quantum noise across different platforms (Lines 330 to 334 and lines 368 to 374).

  1. Results are presented very poorly. Fonts in the figures are too small to read, the lack of grid makes the values hard to read on the plots, and there is almost no explanation of the results and their relevance.  This aspect should be improved before the resubmission.

  • Your feedback on the presentation of results is well-taken. We have remade, increased font sizes, added grids to all figures of the “Results” and “Noise effects” sections for clarity, and improved the explanation and their significance.

  1. There are some other attempts to use multi-agent approach in quantum games.

For example:

- J.A. Miszczak, L. Pawela, J. Sladkowski, General model for a entanglement-enhanced composed quantum game on a two-dimensional lattice, Fluctuation and Noise Letters, Vol. 13, No. 2 (2014), pp. 1450012

- R. Alonso-Sanz. A cellular automaton implementation of a quantum battle of the sexes game with imperfect information, Quantum Information Processing, Vol. 14, pp. 3639–3659 (2015)

- L. Pawela, Quantum games on evolving random networks, Physica A: Statistical Mechanics and its Applications, Vol. 458, pp. 179-188 (2016)

What is (if any) relation of the presented research with the mentioned references? Comments in this regard would be required.

  • We appreciate your references to related work. In the revised manuscript, we extensively described the relationship, distinctions, and contributions of our research in comparison to the mentioned references on multi-agent quantum games in lines 64 to 89.

  1. No source code for the perforce numerical experiments is provided. This makes the research unrepeatable, and hence not very useful for further investigation. The paper cannot be published without providing source code and scripts used to obtain the presented results.

  • We understand the importance of replicability and transparency. A github repository was shared with all the code needed to reproduce all the information reported in the article.

Round 2

Reviewer 2 Report

The revised manuscript is fine for me to be accepted for publication. 

Reviewer 3 Report

The authors included necessary amendments. I suggest accepting the paper in the present form.